# Physical Performance of Brazilian Military Policemen: A Longitudinal Analysis by Occupational Specialties

**DOI:** 10.3390/ijerph192416948

**Published:** 2022-12-16

**Authors:** Luiz Rezende, Rodolfo A. Dellagrana, Luiz Gustavo Rodrigues Oliveira-Santos, Arthur Duarte Fantesia Costa Cruz, Maycon Felipe da Silva Mota, Christianne F. Coelho-Ravagnani

**Affiliations:** 1Research in Exercise and Nutrition in Health and Sports Performance—PENSARE, Faculty of Education (FAED), Post-Graduate Program in Movement Sciences, Institute of Health (INISA), Federal University of Mato Grosso do Sul, Campo Grande 79070-900, Brazil; 2Physical Education Department, State University of Ponta Grossa, Ponta Grossa 840030-900, Brazil; 3Federal University of Mato Grosso do Sul, Campo Grande 79070-900, Brazil; 4School of Public Health Dr. Jorge David Nasser, State Health Department of Mato Grosso do Sul, Campo Grande 79074-460, Brazil

**Keywords:** military police, occupational health, acting, aerobic conditioning, localized muscle strength, physical activity, physical assessment

## Abstract

Maintenance of good levels of physical fitness is essential during occupational tasks for the general health of the military police. However, no studies have evaluated longitudinal changes in the physical fitness of Brazilian military police officers according to their specialties. Thus, the objective of the current study was to analyze the changes in the physical fitness of military police officers according to their specialty, over a period of five years. Retrospective data (2015–2019) from 290 police officers were analyzed, including age and physical fitness tests (12-min run test, sit-ups, push-ups, and pull-ups on the bar). The sample was divided into four groups (Specialized; Border; Urban; and Environmental). ANCOVA was used to describe differences in physical fitness components between groups of police officers after adjusting for age. Initial fitness was higher among police officers in the Specialized group (i.e., those with greater physical demands). During the five-year follow-up period, there was an age-related decrease in physical capacity for all groups, regardless of specialty. However, Urban police showed improvement in running and sit-up tests (*p* < 0.05) over time. Changes in physical capacity during follow-up differed depending on the physical component analyzed and the occupational specialty.

## 1. Introduction

Brazil occupies one of the worst positions in terms of crime in the world [1]. This hostile scenario requires that public safety officers are physically and mentally prepared to deal with high-risk situations, with little time to make decisions [2,3].

The ability of police officers to run, jump, roll, climb, crawl, creep, carry, fight, lift, and transport the weight of equipment and possible injured companions or victims presupposes good levels of physical fitness in cardiorespiratory, flexibility, strength, and muscular endurance components, therefore representing an essential condition for the performance of the functions of the position [4,5]. In addition, higher levels of physical fitness are associated with a lower risk of disease (e.g., obesity, diabetes mellitus, and arterial hypertension) and injury among military police (MP) [6]. On the other hand, low physical fitness can impair the effectiveness of the service, putting the life of the officer, work partners, and the community at risk [4].

Although the literature on the subject points to the importance of maintaining physical fitness throughout the police career, at the same time, reports show a tendency to a decline in the physical capacity of military police officers (MPs) over the years [7,8]. This decline may be related, among other factors, to the function performed by these individuals [9,10,11].

The Brazilian Federal Constitution of 1988 [12], Art. 144, attributes to the Military Police of the States the ostensible policing and preservation of public order. Within the State of Mato Grosso do Sul, the Military Police fulfills this attribution and is institutionally organized so that some units, in addition to the constitutional attributions, also act in specific activities, according to their specialty/attribution, with the main units being: Environmental Military Police Battalion (BPMA); Military Traffic Police Battalion (BPTran); Military Road Police Battalion (BPMRv); Riot Police Battalion (BPChoq); Special Police Operations Battalion (BOPE); Department of Border Operations (DOF); and 1st Military Police Battalion (1ºBPM) [13].

A common requirement of the different specialties of the Brazilian Military Police is the completion of the Physical Fitness Test (PFT), both for admission and for functional promotion or participation in specialization courses. The PFT data could be very useful for understanding how changes in the physical condition of MPs occur and how these changes can affect not only their ability to perform their work activities but also their health. Therefore, understanding the physical fitness behavior of police officers over the years is critical in this population.

In the current literature, and especially in Brazil, no studies were found with longitudinal analyses of the physical fitness of MP officers. In addition, few studies have been conducted with respect to differences between MP occupational specialties. Thus, the objective of the current study was to analyze the physical fitness of MPs of different specialties over five years. We hypothesized that the physical fitness of MPs decreases over the years, as observed in other studies [7,8], but with different patterns of reduction between specialties.

## 2. Materials and Methods

### 2.1. Participants

A retrospective longitudinal study was carried out with an analysis of secondary data from July to October 2021. The research included the results obtained in the biannual PFT carried out by the MPs in their respective units during the period from 2015 to 2019. All participants were male, belonged to the selected units, and had performed at least one of the biannual PFT per year for a period of 5 years. MPs who did not perform the PFT or who had incomplete data in the proposed period for any reason were excluded from the study.

Participants were allocated into four groups determined according to specialty/attribution affinities, as follows: Specialized (*n* = 98; 33.4 ± 4.5 years), containing the MPs assigned to the Special Police Operations Battalion (BOPE) and the Riot Police Battalion (BPChq); Border (*n* = 52; 36.6 ± 4.3 years), containing the MPs assigned to the Military Highway Police Battalion (BPMRv) and the Department of Border Operations (DOF); Urban (*n* = 27; 33.7 ± 5 years), containing the MPs assigned to the 1st Military Police Battalion (1st BPM) and Military Traffic Police Battalion (BPTran); and Environmental (*n* = 113; 41.6 ± 5.7 years), containing the MPs assigned to the Environmental Military Police Battalion (BPMA).

### 2.2. Procedures

After receiving authorization from the head of the Military Police Corporation of Mato Grosso do Sul, the PFT data were obtained from the institution’s specific electronic files, published in the General Command Bulletin (BCG). A specific data coding mechanism was used to guarantee the confidentiality and anonymity of the participants, and access to all data collected for the research was restricted to the researcher and the Military Police Commander. In this way, no individual was personally approached, recruited, or involved in any intervention.

The research was carried out in accordance with the conditions of the Declaration of Helsinki and approved by the Research Ethics Committee (CEP) of the Federal University of Mato Grosso do Sul (UFMS) with a favorable opinion according to Opinion No. 4,823,310 of 2 July 2021.

### 2.3. Physical Fitness Assessment

The variables age and the PFT results were used, which included: 12-min run; pull-ups (Exercises on the Bar); push-ups; and sit-ups. The protocols for performing the tests that make up the PFT battery are described in the PMMS Ordinance No. 042/PM-1/EMG/2018, of 20 August 2018, published in BCG No. 156 of 21 August 2018 [14]. The exercises that make up the PFT are all performed on the same day. The sequence of exercises begins with the Pull-ups on the bar or Push-ups on the floor, followed by the Sit-ups, and 12-Minute Run. At the discretion of the evaluators, the tests are started, without a defined interval between the first two tests (Pull-ups and Push-ups) and with an interval of approximately 30 min before the start of the Running Test, after the first two tests have been performed. The warm-up was controlled by the participant and involved low-intensity running and racing.

Pull-ups (exercises on the bar): The MPs started from the initial position with the grip in pronation and arms outstretched. The complete execution of the exercise began with the grip of the hands on the bar, with the performer keeping their arms and legs fully extended and performing flexion of the arms until passing their chin line over the bar and returning to the starting position. The position of the legs is free, and the height of the bar is adjusted according to the candidate’s height so that the candidate does not touch the ground when their arms are extended above their body. The highest number of repetitions was performed, in an uninterrupted manner, until fatigue or failure to perform the technique correctly. An official inspector accompanied the test to carry out the count [15].Push-ups: The MPS started from the position with the body extended horizontally on the ground with arms extended perpendicular to the shoulders and hands flat on the ground. Flexion of the arms was performed to form a ninety-degree angle so that the elbow was aligned with the back. One repetition was counted starting from the extended arms position, executing the arm flexion movement, and returning to the initial position. The highest number of repetitions was counted uninterruptedly until fatigue or failure to perform the technique correctly. An inspector accompanied the test to carry out the count [15].Sit-ups: The MPs started from the initial position lying in dorsal decubitus (back on the floor), arms crossed on the chest, hands resting on the shoulders, knees bent, feet resting on the floor and fixed to a support point) which could be a bar, or even the official inspector or someone else, heels close to the buttocks. They performed abdominal flexion, removing their shoulder blades from the ground, until the elbows made contact with the knees or quadriceps, then returned to the initial position until they touched the ground with the upper half of the shoulder blade, performing the highest number of repetitions uninterruptedly in the exercise until fatigue or failure to perform the technique correctly. An official inspector accompanied the test to carry out the count [15].The 12 min run: On an official athletics track of 400 m (one lap), identification cones were placed every 50 m. The test lasted 12 uninterrupted minutes; at 11 min a whistle was blown to warn the MPs about the time, and at 12 min the final whistle was blown, at which time the police officers immediately stopped running. The distance covered was measured in meters. The time was monitored using a stopwatch device. All tests were accompanied by an official inspector (PMMS, 2018). To calculate the VO_2_max, the following formula by Cooper [14] was used: VO_2_max (mL/kg/min) = Distance covered (meters)—504/45.

### 2.4. Statistical Analysis

Descriptive statistics of mean and standard deviation were used for the presentation of age and results of physical tests. To compare the results of the physical tests between the police specialties at baseline and throughout the study periods (five years), Analysis of Covariance (ANCOVA) of a General Linear Mixed Model was performed, controlled for age differences between groups, followed by Tukey’s post-hoc, using the nlme package within the R program. Data normality was verified using the Shapiro–Wilk test. The police specialty group (categorical variable), time (continuous variable), and the police officer’s age (continuous variable) were used as fixed variables and the police officer’s identity as a random intercept variable. Interaction between groups and time was enabled to verify potential differences in changes of direction in performance over time, depending on the investigated group. The inclusion of the police officer’s identity as a random variable in the model aimed to simulate a model of repeated measures, allowing general conclusions to be drawn by longitudinally following the performance of each participant over time. The environmental group was adopted as a reference group, and age control was used for performance analysis. The level of statistical significance was set at *p* < 0.05 for all analyses.

## 3. Results

The sample consisted of 290 male police officers allocated into 4 (four) different units of the corporation, comprising the groups: Environmental (*n* = 113), Specialized (*n* = 98), Border (*n* = 52), and Urban (*n* = 27). Table 1 shows the descriptive data of the physical tests according to police occupations and respective years of study.

Figure 1 shows the age distribution of police officers according to their respective specialties.

The results of the 12-min run test, using the Environmental group as a reference, showed that at baseline (Figure 2A; Table 2) the specialized group demonstrated higher performance (βspecialized = 173.28 m, *p* < 0.05). On the other hand, the Border and Urban groups presented performances statistically equivalent to the Environmental group (βborder = −37.47 m, *p* > 0.05; βurban = 58.57 m, *p* > 0.05). The Environmental group (reference) showed an annual decline of −55.16 m in the distance traveled (*p* < 0.05) (Table 2). There was an interaction with all groups, and in relation to the reference group, the Specialized group showed a smaller decline (−35.49 m; βspecialized:time = 19.67; *p* < 0.05), while the Border group presented an equivalent decline (−44.17 m; βborder:time = 10.99; *p* > 0.05) and the Urban group showed a significant increase in the distance traveled (+16.37 m; βurban:time = 71.53; *p* < 0.05) over the five years. Regarding the effect of age, there was a significant decline of −7.81 m per year in the distance covered by the police officers (*p* < 0.05) (Figure 2B). The post-hoc comparison between the groups starting from the baseline behaved as follows (Figure 2C): Environmental ≠ Specialized, Specialized ≠ Border, and Border ≠ Urban.

With regard to VO_2_max, the results show that at baseline (Figure 3A; Table 3) the Specialized group demonstrated higher performance than the reference group (Environmental) (βspecialized = 3.85 mL/kg/min, *p* < 0.05), the Border and Urban groups presented statistically equivalent performances (βborder= −0.83 mL/kg/min, *p* > 0.05; βurban = 1.30 mL/kg/min, *p* > 0.05). The Environmental group (reference) showed an annual decline of –1.22 mL/kg/min in VO_2_max (*p* < 0.05). There was an interaction with all groups, and in relation to the reference group, the Specialized group showed a lower decline (−0.79 mL/kg/min; βspecialized:time = 0.43; *p* < 0.05), while the Border group showed an equivalent decline (−0.98 mL/kg/min; βborder:time = 0.24; *p* > 0.05) and the Urban group showed a significant increase in VO_2_max (+0.36 mL/kg/min; βurban:time = 1.58; *p* < 0.05) over the five years. Regarding the effect of age, there was a significant decline of −0.17 mL/kg/min per year in VO_2_max (*p* < 0.05) (Figure 3B). The behavior of the post-hoc comparison between the groups starting from baseline was as follows (Figure 3C): Environmental ≠ Specialized, Specialized ≠ Border, and Border ≠ Urban.

For the push-ups test, the results at baseline (Figure 4A; Table 4) indicated that the Specialized group again demonstrated a significantly higher performance (βspecialized= 5.74; *p* < 0.05), while the Border and Urban groups presented performances equivalent to the reference group (Environmental) (βborder = −2.01; *p* > 0.05); (βurban = 3.82; *p* > 0.05. There was a significant annual decline of −1.36 repetitions in the push-up test (*p* < 0.05) for the reference group. There was interaction with all groups, and the Specialized and Urban groups showed declines equivalent to those of the reference group, without significant differences −1.9 repetitions; βspecialized:time = −0.54; *p* > 0.05); −0.67 repetitions; βurban:time = 0.69; *p* > 0.05) and the Border group showed an increase in performance, with a significant difference of 0.15 repetitions; βborder:time = 1.51; *p* > 0.05). Regarding age, a significant decline of −0.26 repetitions per elapsed year (*p* < 0.05) was observed (Figure 4B). The behavior of the post-hoc comparison between the groups starting from the baseline was as follows (Figure 4C): Environmental ≠ Specialized.

The performance in the pull-ups test showed similar behavior to the other tests at baseline, where the Specialized group was the only one that presented a statistically significant superior performance, (βspecialized = 2.63; *p* < 0.05), while the Border and Urban groups presented performances equivalent to the reference group (Environmental) (βborder = 0.80; *p* > 0.05 and βurban = 0.69; *p* > 0.05), (Figure 5; Table 5). The reference group did not show a significant annual decline in the number of repetitions performed in the test (β = −0.20 *p* > 0.05). There was interaction with all groups, with the Specialized group showing a significant decline in performance (−0.63 βspecialized:time = −0.43; *p* < 0.05), and the Border and Urban groups remained equivalent to the performance of the Environmental group, without significant differences (−0.63; βborder:time = −0.43; *p* > 0.05); (−0.35; βurban:time = −0.15; *p* > 0.05). Regarding the effect of age, there was a significant decline of −0.11 repetitions performed by police officers per year (*p* < 0.05) (Figure 5B). The post-hoc comparison behavior between the groups starting from the baseline was as follows (Figure 5C): Environmental ≠ Specialized; Specialized ≠ Urban; Specialized ≠ Border.

Figure 6 and Table 6 present the statistical results for the Sit-ups test. It can be observed that at baseline the Specialized group again presented better performance (βspecialized = 7.64; *p* < 0.05), while the Urban group presented lower performance (βurban = −4.00; *p* < 0.05) and the Border group presented performance equivalent to the reference Environmental group (βborder = −2.06; *p*> 0.05). A significant annual decline in the number of repetitions performed in the test (β = −2.08; *p* < 0.05) was observed for the Environmental group. There was interaction with all groups, and the Specialized group showed a significant decline of −2.73 repetitions (βspecialized:time = −0.65; *p* < 0.05), while the Urban group showed a significant improvement in performance + 0.47 repetitions; (βurban:time = 2.55; *p* < 0.05), and the Border group remained equivalent to the reference group, with no significant difference, with −1.55 repetition; (βborder:time = 0.53; *p* > 0.05). Considering age, the results show that with each passing year, police officers perform 0.56 fewer repetitions (*p* < 0.05) (Figure 6B). The post-hoc comparison between the groups starting from the baseline behaved as follows (Figure 6C): Environmental ≠ Specialized; Environmental ≠ Urban; Specialized ≠ Urban; Specialized ≠ Border.

## 4. Discussion

This is the first retrospective longitudinal study to investigate the physical fitness of Brazilian Military Police from different units. The results of this study indicated that, in general, there was a decrease in the physical fitness of MPs with advancing age. However, changes in physical capacity throughout the follow-up differed depending on the physical component analyzed and the police specialty. It was observed that police officers with greater physical demands—i.e., the specialized group—showed better physical fitness in all components in the baseline year, while environmental, urban, and border groups presented equivalent performances. Cardiorespiratory and abdominal endurance were the most affected components of physical fitness throughout the follow-up.

The MPs assigned to the BOPE and BPChq (Specialized group) have greater physical and technical demands in their missions/operations and physical fitness is an essential condition for these police officers to carry out their duties, to preserve not only their own lives during the operations but also those of their battalion colleagues. Therefore, maintenance of physical training and the pursuit of specialization are requirements for the performance and permanence of these police officers in the specialized unit [2,16] and may explain their superiority in physical fitness in relation to the other police units analyzed in the present study.

In the same way, the lower levels of cardiorespiratory and muscular resistance in urban, environmental, and border police officers when compared to the specialized officers in our study may be related to the nature of the routine performed by these groups of police officers. The MPs of the environmental group generally carry out operations with less physical involvement, as their main function has always been to establish a set of norms and instruments aimed at minimizing the negative impacts of human action in relation to the environment. Thus, as they carry out more administrative work, BPMA MPs spend most of their workday performing actions that do not require much physical effort, and when they are on missions, they use vehicles for transportation. The work activities carried out by the border group also involve long periods of sitting during the journey, especially the officer driving the vehicle on patrol through the border regions. These MPs also often work in shifts.

In the current study, the urban group included MPs from the 1st BPM and BPTran. The former act mainly in ostensive, preventive policing in the city, that is, they perform training, that is not superior to that of the specialized group. BPTran MPs, on the other hand, carry out more administrative work, as there is no requirement for high physical fitness to perform their duties. This ambiguity between the 1st BPM and BPTran may have contributed to the relatively lower performance compared to the specialist group. However, it is worth mentioning that despite the physical superiority of the specialized group at baseline, this was the only group to show a decline in all tests performed during follow-up. A possible explanation lies in the fact that specialized MPs start from higher levels of physical fitness and therefore tend to decline further. Individuals who are more trained and with higher fitness levels have less potential for gains than those with lower fitness levels. In the same way, fitter individuals are expected to show a greater decline [17].

This hypothesis is reinforced if we compare the physical fitness of the specialized police officers in the present study with that of other Brazilian police officers. Pereira and Teixeira [7] set out to establish normative values, considering the age in physical tests of 985 Brazilian male military personnel from the Air Force. The average number of push-ups on the floor performed until fatigue was 22.03 ± 7.47 repetitions and the average distance covered in the 12-min running test was 2485.30 ± 322.42 m. BOPE police officers in Santa Maria/RS performed an average of 24.19 ± 9 push-ups and ran 2298 ± 352 m [16]. In both studies, the values were lower than those reached by the specialized MPs in the current study, who performed an average of 33 push-ups on the ground and covered an average distance in the 12-min run of 2599 m, showing the high physical conditioning of this group. Corroborating our findings, the authors observed that physical performance tends to decrease with advancing age in all analyzed tests [7].

Similar to what was observed in the specialized group, significant declines in cardiorespiratory and abdominal resistance were also observed in police officers in the border and environmental groups, showing these components of physical fitness as the most affected in the current study. These findings, although expected, are worrisome, as lower levels of cardiorespiratory fitness are associated with worse health indicators [18], higher levels of adiposity [19,20,21], and worse work ability [5] among police officers. Additionally, good abdominal strength is associated with faster times in the 90-m obstacle course (r = −0.208), 1.8-m wire fence climbing (r = −0.175), and 457-m run (r = −0.344) and therefore with the ability to perform the tasks required in patrol work [22].

Contrary to the results observed for the aforementioned police groups, in our study, only the MPs from the urban group showed improvement in aerobic physical fitness and abdominal resistance during the follow-up period. In addition, there was the maintenance of the values in the pull-up test. These findings are possibly due to the fact that newly graduated police officers or those undergoing training are usually assigned to ostensive and preventive policing battalions, for example, the 1st BPM (urban group), with the aim of gaining experience in policing/urban patrol. In general, they are younger police officers, in training and in the evaluation process, who need to maintain good physical fitness to be positively evaluated. Another interesting point is the fact that, as they are novice police officers, they aim to advance professionally to other groups, such as the specialized group. In addition, during the evaluation process, only the best placed in the training process can choose which unit they want to work in.

Previous studies show that newly graduated recruits from the academy tend to present better physical fitness with better performance in strength, endurance, muscle power, and aerobic capacity compared to previously graduated police officers [23,24]. In this sense, Orr et al. [10] compared US recruits and rangers and found that rangers took 16% longer to complete a 2.4 km run and performed 18% fewer push-ups and 15% fewer sit-ups in 60 s compared to recruits.

Our findings, therefore, partially corroborate those of Matos et al. [25], where 31 MPs from the state of Rio de Janeiro (PMERJ) were evaluated, aged between 30 and 39 years, during four years of PFT (2007–2010). The authors found no decline in sit-up and pull-up tests over time. In 2007 the MPs performed an average of 39 sit-ups and in 2010, 40 sit-ups, while in the pull-ups on the fixed bar, 4 pull-ups were performed in 2007 and 6.5 in 2010. The distances covered by the MPs in the 12-min test were 2506 m, 2645 m, 2648 m, and 2581.33 m in 2007, 2008, 2009, and 2010, respectively, with a significant decline (*p* = 0.003) only from 2009 to 2010.

Interestingly, in our study, the push-up test was the least affected over the four years. With the exception of the specialized group, all other police specialties showed maintenance of performance in this test. Similar results were observed by Boyce et al. [26] who longitudinally evaluated the physical fitness of police officers through batteries of physical tests. The authors observed gains in strength and/or maintenance in strength/kg of upper limbs in the bench press after 12.5 years of police service, which were probably associated with physical training (since the group was formed by recruits in training for the police), in addition to the biological maturation process itself (as they were young police officers) and lean mass gains. It must also be considered that according to the norms of the Military Police of Mato Grosso do Sul, military police officers aged 36 years or older have the right to choose between carrying out the push-up test on the ground or the pull-up test on the bar [15]. Thus, it is assumed that because it is a test with a greater degree of difficulty, only the police officers who considered themselves more apt chose to perform pull-ups on the bar, a fact that could explain our findings.

It should be noted that in the present study, age was a key factor that contributed to the reduction in the physical fitness of police officers. These findings were also systematically observed in other studies [23,27,28]. Teixeira et al. [27], analyzed physical fitness performance using seven tests (palm grip, vertical jump, long jump, sit-ups, push-up, bench press test, and simulation test with climbing, shooting, etc.) of 97 male police officers in Lisbon/Portugal. Through multiple regression analysis, the authors highlighted that there was a drop in physical performance (long jump, abdominal, aerobic capacity, and simulation test) in police officers in the category > 49 years compared to those aged 20–29 years.

In the same sense, Sorensen et al. [28] showed that the muscular performance of Finnish policemen decreased with years of service. Lagestad et al. [8] followed Norwegian male police officers for 16 years in four tests (bench press, pull-up bar, long jump, and 3000 m sprint). Officers’ fitness levels decreased by approximately 10–32% in all physical tests. In another work, Dawes et al. [29] observed a decline in performance in the vertical jump test, floor push-ups, sit-ups, and shuttle run tests in the 30–39, 40–49, and 50–59 age groups, compared to the younger 20–29 age group of highway police officers. Together, these and our findings demonstrate that physical fitness declines over the years.

The advancement in age itself is related to a series of physiological and morphological alterations, such as increases in the percentage of fat and BMI [30], and decreases in muscle mass [31], bone density [32], and testosterone levels [33], which can contribute to the reduction in the physical fitness of police officers.

Added to this is the fact that the stressful and sedentary nature of police work, with occasional periods of maximum or near-maximum physical exertion [34,35] predisposes this population to a higher risk of injury and morbidity [36] and mortality compared to the general population [24,34]. Other lifestyle-related factors that are known to compromise physical fitness, such as physical inactivity, poor diet, sleep disorders, and alcohol and tobacco use, are highly prevalent among Brazilian police officers [19,37,38] similar to other countries such as India [36], the United States [5,39], Serbia [11], and Finland [28]. Taken together, these factors could explain the age-related decline in the physical fitness of the police officers in our study.

### Limitations and Future Perspectives

Some limitations can be pointed out. First, we used the documents of the Military Police of Mato Grosso do Sul as the main source of data, containing the results of the PFT. However, we do not have anthropometric and health data that could contribute to the discussion in this work. It is suggested that these data be included in future studies. Another limitation is the fact that PFT evaluations are prone to intra and inter-evaluator errors, since they are carried out by different teams, in different places, and under different weather conditions. In addition, the generalization of their data is limited, as the athletes come from a single region of the country. Considering also the average age of police officers, it is likely that most of them are still in the first half of their careers. Thus, further longitudinal studies involving a representative sample of the country are necessary to assess several factors that may contribute to the physical fitness, injury records, and general health of these agents throughout their professional trajectories, in order to improve their occupational conditions.

Despite the limitations, our study demonstrated the need to pay attention to the physical fitness of the police in addition to the performance of the PFT. It is necessary to consider the need to implement physical conditioning programs within the battalions themselves, especially for longer-serving, and older police officers, in order to preserve their health, physical fitness, and work capacity. The practice of physical activity should be worked on within an integral context and aimed at health promotion, not only as a necessary rule for the development of the function, even if it indicates the good performance of these officers in their occupation [39].

## 5. Conclusions

The present study showed that the physical fitness of police officers declines with advancing age, but the longitudinal changes behave differently, relating to the nature of the police activity. These findings raise the importance of adopting measures to monitor and maintain or improve the physical capacity of police officers, given their relevance to health and occupational activity, reflecting on positive results for society in general.

## Figures and Tables

**Figure 1 ijerph-19-16948-f001:**
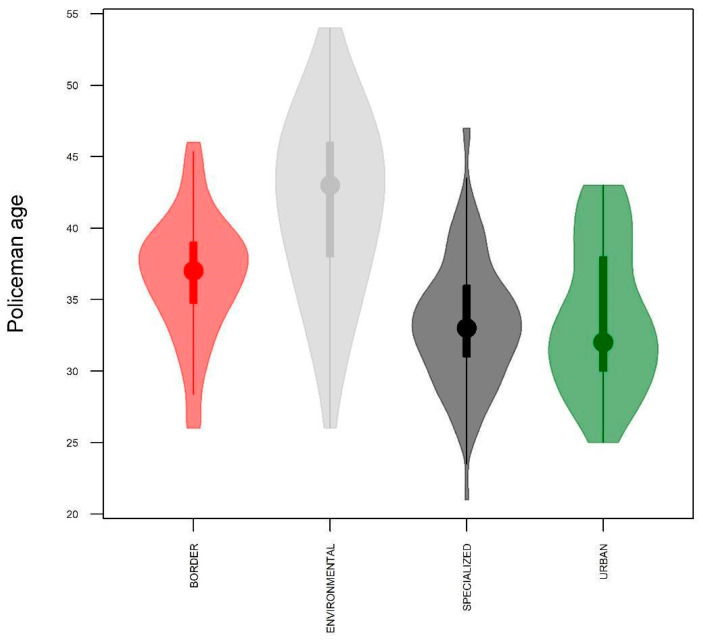
Age distribution of police officers according to specialties.

**Figure 2 ijerph-19-16948-f002:**
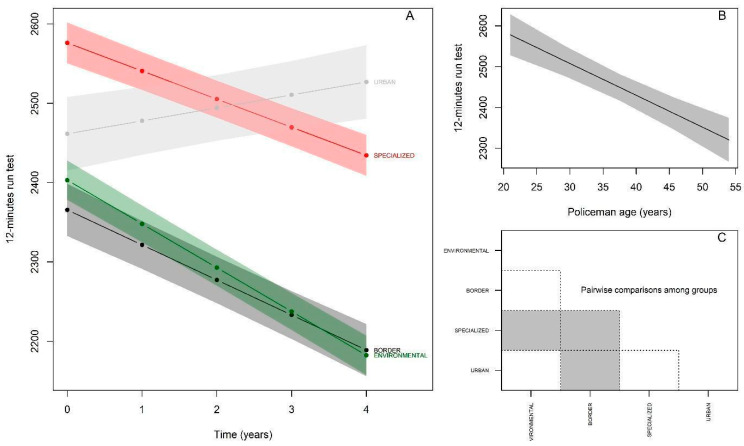
Behavior of the performance of police officers in different specialties in the 12-min run test throughout the follow-up. (**A**): Changes in distances covered over 5 years according to police specialties; (**B**): Distance traveled by police in relation to age; (**C**): Comparison of distances covered between police specialties at baseline (gray color indicates significant difference between the respective groups).

**Figure 3 ijerph-19-16948-f003:**
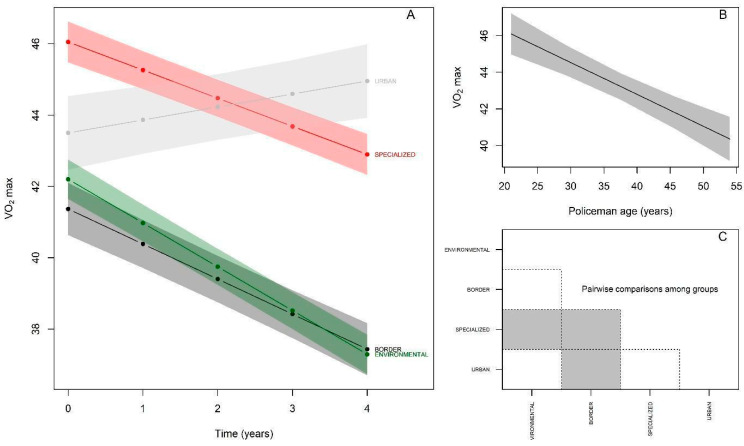
VO_2_max behavior of police officers over the years. (**A**): Changes in VO_2_max performance over 5 years according to police specialties; (**B**): VO_2_max performance achieved by police officers in relation to age; (**C**): Comparison of VO_2_max achieved between police specialties at baseline (gray color indicates significant difference between the respective groups).

**Figure 4 ijerph-19-16948-f004:**
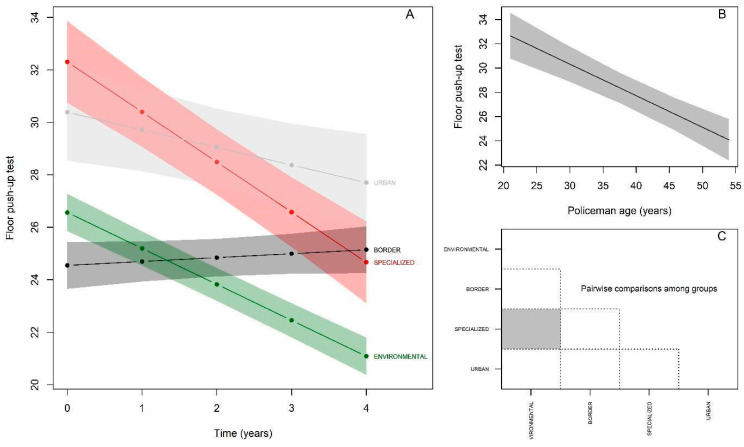
Performance behavior of police officers—Push-ups. (**A**): Change in maximal floor push-up repetition performance over 5 years by police specialties; (**B**): Performance in maximal push-up repetitions by police officers in relation to age; (**C**): Comparison of test performance between police specialties at baseline (gray color indicates significant difference between respective groups).

**Figure 5 ijerph-19-16948-f005:**
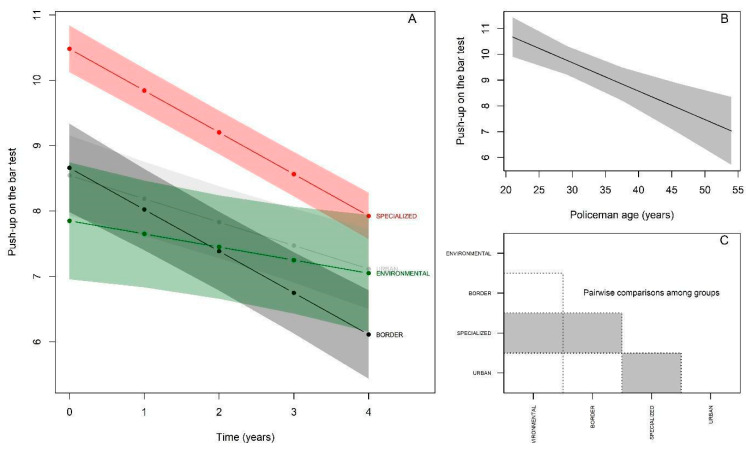
Performance behavior of police officers—Pull-ups. (**A**): Change in performance of maximum repetitions of pull-ups on the bar over 5 years according to police specialties; (**B**): Performance in maximum repetitions of pull-ups on the bar by police officers in relation to age; (**C**): Comparison of performance on the push-up test between police specialties at baseline (gray color indicates significant difference between the respective groups).

**Figure 6 ijerph-19-16948-f006:**
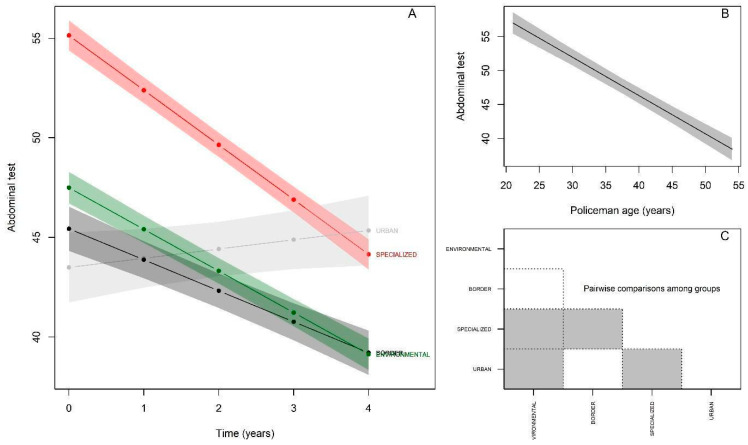
Performance behavior of police officers—Sit-ups Test. (**A**): Change in performance of maximum repetitions of sit-ups over 5 years according to police specialties; (**B**): Performance in maximum repetitions of sit-ups by police officers in relation to age; (**C**): Comparison of sit-up test performance between police specialties at baseline (gray color indicates significant difference between respective groups).

**Table 1 ijerph-19-16948-t001:** Descriptive values of the performance of military police officers according to the tests.

	2015	2016	2017	2018	2019
	X ± SD	X ± SD	X ± SD	X ± SD	X ± SD
	12-min run (m)				
Environmental (*n* = 113)	2326.8 ± 211.4	2354.8± 251.9	2262.7 ± 276.1	2234.9 ± 296.9	2110.9 ± 270.4
Specialized (*n* = 98)	2599.6 ± 226.5	2568.6 ± 225.5	2558.2 ± 259.6	2486.3 ± 251.7	2463.3 ± 252.9
Border (*n* = 52)	2371.9 ± 283.1	2 339.3 ± 290.3	2272.7 ± 294.5	2206.5 ± 305.4	2217.5 ± 270.8
Urban (*n* = 27)	2475.2 ± 240.2	2537.0 ± 284.1	2485.2 ± 255.6	2573.3 ± 281.5	2538.9 ± 286.7
	VO_2_max (mL/kg/min)				
Environmental (*n* = 113)	40.5 ± 4.7	41.1 ± 5.6	39.1 ± 6.1	38.5 ± 6.6	35.7 ± 6.0
Specialized (*n* = 98)	46.6 ± 5.0	45.9 ± 5.0	45.6 ± 5.8	44.1 ± 5.6	43.5 ± 5.6
Border (*n* = 52)	41.5 ± 6.3	40.8 ± 6.5	39.3 ± 6.5	37.8 ± 6.8	38.1 ± 6.0
Urban (*n* = 27)	43.8 ± 5.3	45.2 ± 6.3	44.0 ± 5.7	46.0 ± 6.3	45.2 ± 6.4
	Push-ups (maximum repetitions)				
Environmental (*n* = 82)	23 ± 4	31 ± 8	22 ± 4	22 ± 4	18 ± 6
Specialized (*n* = 10)	33 ± 10	31 ± 9	28 ± 3	30 ± 11	24 ± 8
Border (*n* = 31)	23 ± 4	25 ± 6	24 ± 5	24 ± 5	24 ± 8
Urban (*n* = 07)	31 ± 8	31 ± 7	26 ± 3	30 ± 9	28 ± 7
	Pull-ups (maximum repetitions)				
Environmental (*n* = 06)	8 ± 2	9 ± 2	8 ± 3	8 ± 3	8 ± 2
Specialized (*n* = 64)	11 ± 2	10 ± 2	10 ± 2	9 ± 3	8 ± 2
Border (*n* = 11)	9 ± 2	9 ± 2	8 ± 2	7 ± 2	7 ± 3
Urban (*n* = 14)	9 ± 2	9 ± 2	8 ± 3	9 ± 2	8 ± 3
	Sit-ups (maximum repetitions)				
Environmental (*n* = 89)	42 ± 7	45 ± 9	43 ± 7	43 ± 9	32 ± 12
Specialized (*n* = 97)	56 ± 8	56 ± 9	53 ± 9	49 ± 7	45 ± 9
Border (*n* = 39)	44 ± 7	44 ± 7	44 ± 6	42 ± 8	37 ± 7
Urban (*n* = 16)	44 ± 11	47 ± 8	46 ± 6	51 ± 7	44 ± 5

Descriptive values (Mean and Standard Deviation).

**Table 2 ijerph-19-16948-t002:** Statistical summary and model-estimated coefficients of the effect of group, time, and age on running performance.

	Estimated Coefficient	Standard Error	Degrees of Freedom	t-Value	*p*-Value
Reference group (Environmental)	2693.84	105.91	1156	25.43	0.00
Specialized	173.28	38.91	285	4.489	0.00
Border	−37.47	41.58	285	0.90	0.36
Urban	58.57	54.45	285	1.07	0.28
Time effect (follow-up)	−55.16	5.09	1156	−10.83	0.00
Age effect	−7.81	2.48	285	−3.14	0.00
Environmental vs. Specialized	19.67	7.46	1156	2.63	0.00
Environmental vs. Border	10.99	9.06	1156	1.21	0.22
Environmental vs. Urban	71.53	11.59	1156	6.17	0.00

**Table 3 ijerph-19-16948-t003:** Statistical summary and model-estimated coefficients of the effect of group, time, and age on VO2max performance.

	Estimated Coefficient	Standard Error	Degrees of Freedom	t-Value	*p*-Value
Reference group (Environmental)	48.66	2.35	1156	20.67	0.00
Specialized	3.85	0.85	285	4.49	0.00
Border	−0.83	0.92	285	−0.90	0.36
Urban	1.30	1.21	285	1.07	0.28
Time effect (follow-up)	−1.22	0.11	1156	−10.83	0.00
Age effect	−0.17	0.05	285	−3.14	0.00
Environmental vs. Specialized	0.43	0.16	1156	2.63	0.00
Environmental vs. Border	0.24	0.20	1156	1.21	0.22
Environmental vs. Urban	1.58	0.25	1156	6.17	0.00

**Table 4 ijerph-19-16948-t004:** Statistical summary and model-estimated coefficients of the effect of group, time, and age on performance in the floor push-up test.

	Estimated Coefficient	Standard Error	Degrees of Freedom	t-Value	*p*-Value
Reference group (Environmental)	36.24	3.36	516	10.75	0.00
Specialized	5.74	1.74	125	3.28	0.00
Border	−2.01	1.07	125	−1.86	0.06
Urban	3.82	1.99	125	1.91	0.05
Time effect (follow-up)	−1.36	0.163	516	−8.36	0.00
Age effect	−0.26	0.07	125	−3.37	0.00
Environmental vs. Specialized	−0.54	0.49	516	−1.09	0.27
Environmental vs. Border	1.51	0.31	516	4.85	0.00
Environmental vs. Urban	0.69	0.58	516	1.19	0.23

**Table 5 ijerph-19-16948-t005:** Statistical summary and model-estimated coefficients of the effect of group, time, and age on performance in the push-up on the bar.

	Estimated Coefficient	Standard Error	Degrees of Freedom	t-Value	*p*-Value
Reference group (Environmental)	11.95	1.71	376	6.96	0.00
Specialized	2.63	0.84	90	3.10	0.00
Border	0.80	0.99	90	0.81	0.41
Urban	0.69	0.95	90	0.72	0.47
Time effect (follow-up)	−0.20	0.20	376	−0.95	0.33
Age effect	−0.11	0.05	90	−2.14	0.03
Environmental vs. Specialized	−0.43	0.21	376	−2.01	0.04
Environmental vs. Border	−0.43	0.25	376	−1.68	0.09
Environmental vs. Urban	−0.15	0.24	376	−0.62	0.52

**Table 6 ijerph-19-16948-t006:** Statistical summary and model-estimated coefficients of the effect of group, time, and age on performance in the abdominal test.

	Estimated Coefficient	Standard Error	Degrees of Freedom	t-Value	*p*-Value
Reference group (Environmental)	68.44	2.98	960	22.95	0.00
Specialized	7.64	1.16	236	6.58	0.00
Border	−2.06	1.36	236	−1.51	0.13
Urban	−4.00	1.95	236	−2.04	0.04
Time effect (follow-up)	−2.08	0.23	960	−8.95	0.00
Age effect	−0.56	0.06	236	−8.06	0.00
Environmental vs. Specialized	−0.65	0.32	960	−2.03	0.04
Environmental vs. Border	0.53	0.42	960	1.26	0.20
Environmental vs. Urban	2.55	0.59	960	4.26	0.00

## Data Availability

All data are stored in institution-specific electronic files and published in the General Command Bulletin (BCG). Upon request, it was possible to consult these files after authorization from the head of the Military Police Corporation of Mato Grosso do Sul.

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
