# Peer review of "Physical Performance of Brazilian Military Policemen: A Longitudinal Analysis by Occupational Specialties"

_ijerph, 2022, doi:10.3390/ijerph192416948_

Round 1

Reviewer 1 Report

Dear Authors

You have written an interesting paper that is focused on a highly specialised population.

However, some parts need to be addressed for greater clarity of your paper.

The introduction is clear and nicely shows the main rationale of the paper.

I would suggest including 2 more longitudinal studies:

-https://www.tandfonline.com/doi/full/10.1080/00140139.2015.1035760

-https://journals.sagepub.com/doi/abs/10.1350/ijps.2014.16.4.349

In references, you use reference 29 of Lagestad et al. and you use it correctly in lines 480-482. However, in lines 518 to 520, a reference is missing and reference 29 is not correct as this was another study (my second recommended link). Therefore, please add the missing reference.

Participants: in lines 81-88 report the number of participants per group. Also, basic data of these groups are missing - age, height, weight. Also, what was their work experience in years for the particular group? Please add

Physical fitness assessment - Please add the following information:

At what time of the day was the testing performed, in what equipment was it performed, what was the warm-up before the test, what was the sequence of tests and what was the break between tests?

12-minute test - report with what device was the time measured and the final distance (what was the accuracy of measurement - m, cm).

Pull-ups - what was the height of the bar and what was the position of the legs (what instructions were given - straight, bent and crossed, etc). Please report

Results - Table 1 - please round up the means and sd without decimal places for reporting as you can only do a full rep in pushups, pull-ups and sit-ups to be counted. It is ok for your statistic analysis but for reporting please round them up.

Table 2 - why was the environmental group a reference group? What were the criteria? Why not any other group? Elaborate

The discussion is well developed and connects to the current literature. The limitations of the study are well stated and acknowledged.

For further studies, I would also recommend looking at the fitness data and injury records of those participants to perhaps see any interesting connections that could address better physical conditioning for a particular group or particular injury site. With these data, you could perhaps lower the occurrence of occupationally connected injuries.

Overall a well-written study. Some parts still need to be addressed by the authors, however, I don't see any major methodological issues. Therefore, I recommend acceptance after minor revision.

Kind regards

Reviewer 2 Report

First of all, I would like to thank you for the opportunity to review this research. The present research report is quite novel and interesting, however, I think that a few small details need to be improved, which I will mention below:

Firstly, remove the words background, methods, results and conclusions from the abstract. It is not necessary to divide the abstract into these parts.

Even if there are no studies in the present sample, add similar studies in the introduction commenting on the effects obtained. This will help to improve the introduction and provide more context.

At the end of the introduction, add some research questions.

In the material and method section, add a section called "Procedure" where you discuss more extensively how you contacted the different state bodies, where the idea came from, etc.

Also, the discussion is very well done as the results are discussed in an extensive way and related to the subject, however, I have seen that at the end they talk about the limitations of the study. Create a new section entitled Limitations and future perspectives and put there the limitations and future perspectives that may arise from this research.

The bibliography needs to be revised in its entirety, as all the documents are poorly cited.  

Finally, I suggest that the wording be revised, as it is difficult to understand some parts of the study (Introduction and discussion).

Round 2

Reviewer 2 Report

Dear authors. 

All of the proposed comments have been improved. That is why I am approving the publication of the research.

Best regards and congratulations.